# Upper-Bound General Circulation of the Ocean: A Theoretical Exposition

Hsien-Wang Ou 

Lamont-Doherty Earth Observatory, Columbia University, Palisades, NY 10964, USA; hsienou0905@gmail.com

**Abstract:** This paper considers the general ocean circulation (GOC) within the thermodynamical closure of our climate theory, which aims to deduce the generic climate state from first principles. The preceding papers of this theory have reduced planetary fluids to warm/cold masses and determined their bulk properties, which provide prior constraints for the derivation of the upper-bound circulation when the potential vorticity (PV) is homogenized in moving masses. In a companion paper on the general atmosphere circulation (GAC), this upper bound is seen to reproduce the observed prevailing wind, therefore forsaking discordant explanations of the easterly trade winds and the polar jet stream. In this paper on the ocean, we again show that this upper bound may replicate broad features of the observed circulation, including a western-intensified subtropical gyre and a counter-rotating tropical gyre feeding the equatorial undercurrent. Since PV homogenization has short-circuited the wind curl, the Sverdrup dynamics does not need to be the sole progenitor of the western intensification, as commonly perceived. Together with GAC, we posit that PV homogenization provides a unifying dynamical principle of the large-scale planetary circulation, which may be interpreted as the maximum macroscopic motion extractable by microscopic stirring, within the confines of thermal differentiation.

**Keywords:** general ocean circulation; Sverdrup dynamics; potential vorticity homogenization; thermal/dynamical coupling; upper-bound circulation



## 1. Introduction

With the advent of satellite imaging, teeming eddies have emerged as a defining characteristic of the ocean motion field [1], but despite the seemingly random (microscopic) motion, the time-averaged (macroscopic) flow nonetheless exhibits a persistent large-scale structure, which defines the general ocean circulation (GOC, all acronyms are listed in Appendix A) of our inquiry. To limit our scope, however, we are concerned only with the vertical-averaged motion above the main thermocline in an ocean confined between meridional boundaries, therefore excluding the circumpolar current and the deep circulation, as well as the shallow tropical circulation.

Theoretical studies of the large-scale ocean structure have traditionally fallen under the rubrics of wind-driven, buoyancy-driven, and thermocline theories with their differing emphases on thermal and dynamical fields. However, because of the inherent coupling of the two, such divisions are somewhat artificial, and the GOC can be explained only as a single manifestation of the coupled field. This coupling is considerably simplified when the thermal field is condensed into an outcropped thermocline and the macroscopic motion is limited to that of the warm layer—a configuration that is highly discernible in the observed ocean, as attested by the widespread use of reduced-gravity models.

While such layer simplification is justified, the commonly employed laminar dynamics is not, as it suffers from two glaring deficiencies. First, coarse-grid numerical calculations (hence entailing only laminar dynamics) show that a proper account of the meridional overturning circulation (MOC) would push the subtropical front to the basin boundary [2], which thus may not confine the subtropical gyre as observed. Significantly, this shortfall

is removed when eddies are resolved, which suggests that the mid-latitude front is the outcome of an eddying ocean, regardless of the wind force [3]. The second deficiency concerns the laminar vorticity balance, which is seen to be strongly perturbed when eddies are resolved in numerical calculations [4,5]. Specifically, because of the material conservation of the microscopic potential vorticity (PV), eddy mixing acts to homogenize the macroscopic PV, thus impairing the laminar vorticity balance, as discussed in Section 2.

Recognizing the foregoing thermal/dynamical coupling and the central role played by eddies, we consider a GOC couched within the thermodynamical closure of our climate theory [6–8] aiming to derive the generic climate state from first principles. Positing this state as a macroscopic manifestation of a coupled nonequilibrium thermodynamics (NT) system, the theory has reduced the ocean to warm/cold masses and determined their bulk properties, and since their derivation involves only thermodynamics, these bulk properties may be regarded as known for our consideration of the GOC without compromising its causality. Incidentally, the prior constraints include a mid-latitude front, which thus may properly confine the subtropical gyre to remove one significant deficiency noted above.

Besides the prior thermal constraints, our GOC is also subjected to different vorticity balance from that of a laminar regime. Invoking the well-demonstrated PV mixing by eddies (Section 2), we consider the asymptotic limit of homogenized PV, which is seen later to produce an upper-bound GOC. While this upper bound represents only a hypothetical limit, it nonetheless has the following advantages: (a) it enables an analytical solution of closed circulation, free from singularity at the outcrop encountered by laminar dynamics (Section 3.4); (b) it has short-circuited the wind curl, so any resemblance to the observed GOC can only be attributed to PV mixing, not the Sverdrup dynamics; (c) it retains a higher-order vorticity balance than the Sverdrup dynamics, so the remnant of the latter constitutes merely a regular perturbation that would not materially alter the modelled GOC (Section 4).

In a companion paper [9], we applied the same principles to derive the upper-bound general atmosphere circulation (GAC), which is seen to resemble the observed prevailing wind, therefore forsaking previous discordant explanations of the easterly trade winds and the polar jet stream. In the present paper, we shall show that the upper-bound GOC may replicate the observed GOC as well, including a western-intensified subtropical gyre and a counter-rotating tropical gyre feeding the equatorial undercurrent (EUC). This dynamical unification of the two general circulations by PV homogenization suggests that they are more fundamental than heretofore perceived.

Regarding the organization of this paper, we first discuss in Section 2 the thermal and PV configurations of the model, including the prior constraints. Subjected to these constraints, we then derive in Section 3 the upper-bound GOC, which is compared with observations. We provide a renewed perspective on the Sverdrup dynamics in Section 4, and conclude the paper in Section 5.

## 2. Model Configuration

This paper completes this author's climate theory aiming to derive the generic climate state from first principles. The central tenet of the theory is that such a state is a macroscopic manifestation of a NT system because of the inherent turbulent nature of the planetary fluids, so its closure involves maximum entropy production (MEP), a plausible generalization of the second law [10]. Although initial application of MEP involves guesswork [11], recent developments have led to its growing acceptance in climate theories, as reviewed in [12]. In addition, in a recent paper [13], we showed that MEP could be a deductive outcome of the fluctuation theorem, and since the latter is of considerable mathematical rigor and has been tested in laboratory [14,15], this linkage would further strengthen the physical basis of MEP.

Applying MEP in our climate theory, we have reduced the coupled ocean/atmosphere to warm/cold masses and deduced their bulk properties, thus providing prior constraints for the present derivation of the GOC. The model ocean is as sketched in Figure 1, which

consists of warm/cold watermasses separated by an outcropped thermocline, a highly discernible first-order description of the observed ocean. The prior constraints (boxed) are the latitudinal extent ($l$), the reduced gravity ($g'$) and the mean depth ($\bar{h}$) of the warm layer (all symbols and their "standard" values are listed in Appendix A). While derivations of $l$ and $g'$ only involve thermodynamics, $\bar{h}$ is constrained by the mechanical energy balance [8], which is too uncertain to be addressed from first principles [2]; taking $\bar{h}$ as a prior constraint, we need to acknowledge this gap in the model closure. As we shall see, these three parameters are sufficient to uniquely specify the upper-bound GOC, underscoring its considerable robustness.

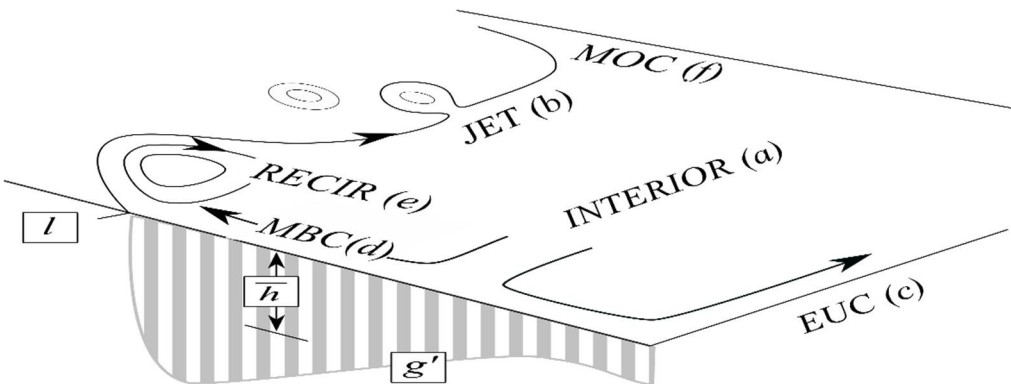

**Figure 1.** The model configuration of a moving warm layer (striped) separated from the motionless cold water by an outcropped thermocline. The prior constraints (boxed parameters) are the latitudinal extent ($l$), the reduced gravity ($g'$) and the mean depth ($\bar{h}$) of the warm layer. The GOC consists of an interior flow (a), the frontal jet (b), the EUC (c), the MBC (d), the recirculation (e) and the MOC (f), which are considered in separate subsections.

Given the differential solar heating, a homogeneous warm layer represents an asymptotic limit of infinite microscopic stirring of the temperature, and since the microscopic PV is materially conserved as the temperature, the macroscopic PV would be similarly homogenized [16]. We shall next assess the relevance of this PV homogenization from the theoretical, computational, and observational perspectives.

From the theoretical perspective, we first note that we use eddy-mixing as a catch-all phrase to include myriad mixing processes, such as turbulent diffusion, shear dispersion, chaotic mixing, mean advection and eddy migration. Among these, chaotic mixing is perhaps the most effective [17], as readily seen in the filamentary appearance of the microscopic PV reflecting its material conservation [18], and it is precisely because of this microscopic conservation that stretching and folding of the filaments would homogenize the macroscopic PV averaged over the filaments. Since filaments are stretched to the boundary before their folding and filling-up of the basin, the macroscopic homogenization proceeds from the basin scale to smaller scales, as seen in the two-particle correlation function [16]. As this time progression is opposite to that of the eddy diffusion, it underscores the inadequacy of the eddy diffusivity in assessing the homogenization process [16], which in principle can be operative even in the absence of eddies [19].

Since the macroscale is defined in its lower limit by the deformation radius, the time required for the two-particle correlation function to equilibrate at this distance provides a tangible measure of the basin-wide homogenization of the macroscopic PV. Moreover, if this "basin mixing time" is short compared with the "basin transit time" of the Sverdrup flow, the latter would not be realized; the contrast between the two timescales thus may differentiate the eddying versus laminar regimes. For a deformation radius of 40 km, the basin mixing time gleaned from [20] (their Figure 17a) is about a decade, which is supported by the tritium-$^3$He age distribution [21]. In comparison, the Sverdrup velocity only reaches several millimeters per second to yield a basin transit time of several decades, so eddy-mixing should dominate the laminar regimes.

For the computational evidence, eddy-resolving numerical calculations have shown palpable homogenization of the PV in the subtropics [4]. For a quantitative assessment, we show in Figure 2 the warm-layer PV profiles adapted from our paper [5] for coarse-grid (1 degree, thick dashed line) and fine-grid (1/32 degree, solid line) calculations representing the laminar and eddying regimes, respectively (the thin dashed lines mark the outcrops). It is seen that the PV range in the eddying regime is reduced to about 20% of its laminar counterpart, which is consistent with the above estimate of the ratio of the basin-mixing to the basin-transit time. This 20% figure may also be interpreted as the "potency factor" retained by the Sverdrup dynamics in refining the upper-bound GOC (Section 4).

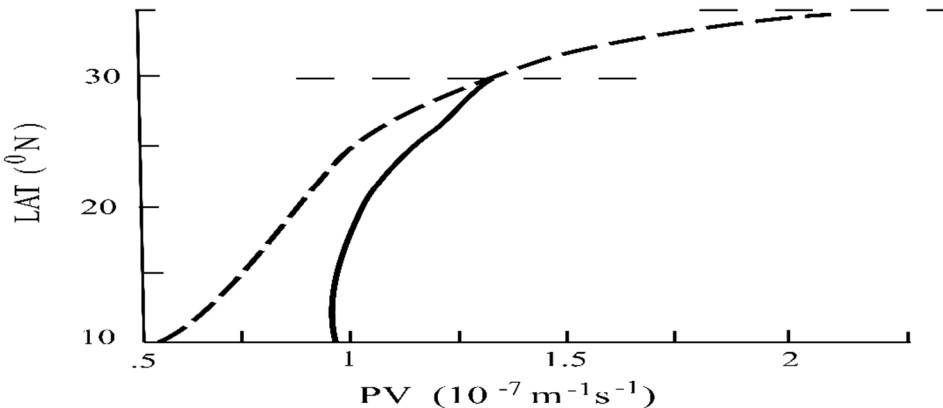

**Figure 2.** Meridional profiles of the warm-layer PV from the coarse-grain (thick dashed) and fine-grain (solid) numerical calculations (adapted from [5]), the thin dashed lines mark the outcrops. The eddy mixing has reduced the PV range to 20% of its laminar range.

We should stress that this PV homogenization is the outcome of strong mixing, and hence differs from that of [22] predicated on weak mixing; the latter is required in their argument to preserve the conservation hence constancy of the PV along a closed streamline, which then diffuses inward to homogenize the PV. In a strong mixing regime, on the other hand, the macroscopic PV is not conserved but smoothed to erase the internal boundary [23], as indeed seen in the tracer distribution [24]. Then, even in a laminar regime, the PV along a closed streamline would be altered as it transits through the frictional boundary layer, nullifying its homogenization [25]. Despite its inapplicability to the observed ocean, the weak mixing argument has nonetheless perpetuated the misconception that the PV is homogenized only in the northwestern corner of the subtropics, which simply does not comport with observations, as discussed next.

For the observational support of the PV homogenization, [26] (his Figure 66) first noticed that the thermocline in the subtropics deepens linearly with the latitude, rendering a columnar PV that is "uniform and nearly constant". Across strong boundary currents where the relative vorticity needs to be included, the columnar PV is again nearly uniform [26] (his Figure 65) and [27] (their Figure 9). The PVs for isopycnal layers have been mapped, but only the ones between $\sigma_\theta$ = 26.5 and 27.0 (below the winter mixed layer and above the main thermocline) are reflective of our columnar PV. From such maps, [28] (their Figure 6) and [29] (their Figure 4a) stated almost identically that "the gyre is dominated by nearly uniform potential vorticity across the *entire* width of the ocean"; similarly, [30] (his Figure 10) noted unequivocally that "in the case of the North Atlantic, the homogenization is very nearly complete". The meridional section [28] (their Figure 15) shows that PV isolines above the main thermocline are generally parallel with isopycnals, indicative of an increasing layer thickness with the Coriolis parameter; this implies the same for their sum, hence the homogenization of the columnar PV. One notes that the degree of PV homogenization should be gauged against its hypothetical distribution in the absence of eddies, such as that shown in Figure 2, which would further accentuate its homogenization. We should express caution, however, that since PV homogenization is a synoptic

feature, it would be degraded in climatological maps [31] due to decadal variability, a well-recognized problem in mapping synoptic water-mass properties [32] (Section 3a), such as the thermocline depth.

To recap, given the shorter basin-mixing time than the basin-transit time by the Sverdrup flow, the PV homogenization above the main thermocline should be quite discernible in the subtropics, as indeed seen in eddy-resolving numerical calculations and observations. Supported by its practical relevance, we are thus justified to consider the asymptotic limit of homogenized PV to derive the upper-bound circulation, as discussed next.

## 3. Upper-Bound GOC

As depicted in Figure 1, the model GOC is divided into its constituents of the interior flow (a), the frontal jet (b), the EUC (c), the meridional boundary currents (MBC, d), recirculation (e) and the MOC (f), as discussed below in successive subsections. Although one may carry out the following derivation in spherical coordinates as in [9], the presence of boundary currents would render the mathematics unwieldy, which we deem as unwarranted given the crudeness of our model. For simplicity, we thus assume a planetary vorticity linear in the latitudinal distance $y$, so the columnar PV ($P$) is

$$P = (\beta y + \xi)/h \tag{1}$$

where $\beta$ is the latitudinal gradient of the Coriolis parameter and $\xi$ the relative vorticity. Since the relative vorticity is negligible in the interior (to be checked later), a homogenized $P$ implies an interior thermocline deepening with latitude, which thus attains a maximum twice its mean depth (before it surfaces across a narrow frontal jet). Defining the scales (bracketed) by $[h] \equiv 2\overline{h}$, $[x, y] = l$, $[u, v] \equiv (g'[h])^{1/2}$, and $[P] = \beta l/[h]$, (1) is nondimensionalized to (all variables are hereafter dimensionless).

$$P = 1 = (y + \varepsilon\xi)/h \tag{2}$$

where $\varepsilon \equiv r_c/l$, with $r_c \equiv (g'[h])^{1/2}(\beta l)^{-1}$ being the deformation radius. In terms of the dynamical closure, we note that the momentum equations and mass continuity have been combined into the PV balance that yields homogenized PV and the geostrophic balance of the macroscopic flow, either in the interior or in boundary layers. Although the boundary layer solutions are generally known, they need to be adapted to accommodate the homogenized PV, and since the boundary layers are one to two orders narrower than the interior, these solutions need to be plotted later in strongly magnified cross-sections.

### 3.1. Interior

Since $\varepsilon \ll 1$ (Appendix A), (2) implies

$$h = y \tag{3}$$

in the interior, as shown in Figure 3. This poleward deepening of the thermocline until its abrupt surfacing across a narrow frontal jet is a well-observed feature [32]. However, it differs qualitatively from that based on Sverdrup dynamics, which has a maximum depth near the maximum wind curl and shoals gradually to the north [33]. Since it is the maximum thermocline depth that enters the velocity scale (Appendix A), the homogenized PV that maximizes this depth produces a GOC that is an upper bound hence the title of the paper.

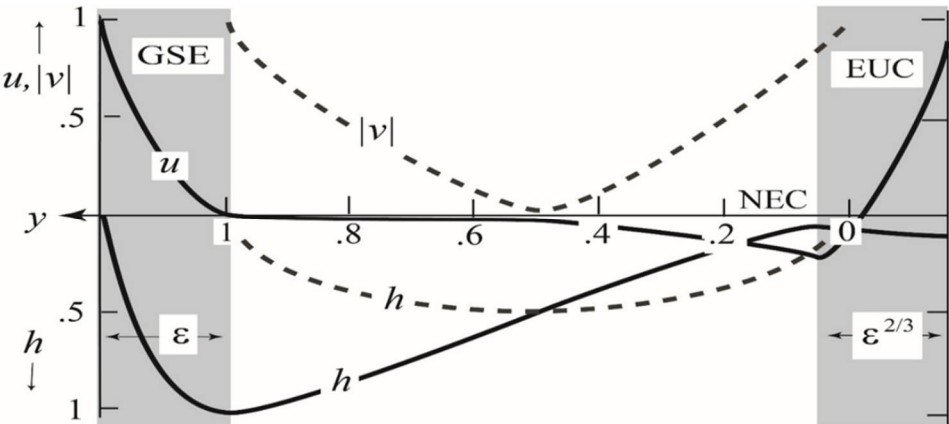

**Figure 3.** The model solution plotted against the latitude. Solid and dashed lines are solutions outside the meridional boundary layers and along the meridional boundaries, respectively. The boundary layers along the subtropical front and equator (shaded) have been magnified 21 and 5 times, respectively.

Assuming the zonal current to be geostrophic, it is given by

$$u = -\varepsilon h_y / y \tag{4}$$

$$= -\varepsilon / y \tag{5}$$

Physically, a deepening thermocline causes a westward geostrophic flow, which intensifies toward low latitudes because of the decreasing Coriolis parameter; the latter can be identified with the north equatorial current (NEC). There is no singularity as $y$ approaches zero since, as we shall see later, there is an equatorial boundary layer where the EUC resides.

Setting the scale of the stream function $[\psi] = [h][u]r_c$, it satisfies

$$\psi_y = -\varepsilon^{-1} h u \tag{6}$$

$$= 1 \tag{7}$$

That is, owing to the opposite latitudinal variation of the thermocline depth and zonal flow, the zonal transport per unit latitudinal distance is uniform and its interior total is unity,

$$[\psi]_0^1 = 1 \tag{8}$$

### 3.2. Frontal Jet

Along the outcrop, there is an eastward geostrophic jet, which can be identified with the Gulf Stream (GS) extension (GSE). The boundary layer solution is well known (for example, [26] (Chapter 8)), but is adapted here to match the interior solution. Defining a stretched coordinate $\varsigma \equiv \varepsilon^{-1}(1-y)$, the homogenized PV (2) and the geostrophic balance (4) state

$$h = 1 + u_\varsigma \tag{9}$$

$$u = h_\varsigma \tag{10}$$

respectively, which yield a solution

$$u = \exp(-\varsigma) \tag{11}$$

$$h = 1 - \exp(-\varsigma) \tag{12}$$

as seen in Figure 3, where the boundary layer width is magnified 21 times (that is, if unmagnified, the front would be aligned with $y = 1$). The transport of the frontal jet can be seen from (10) and (12) to be

$$
\begin{aligned}
[\psi]_0^\infty &= \int_0^\infty hu \, d\varsigma \\
&= \int_0^\infty h h_\varsigma \, d\varsigma \\
&= 1/2
\end{aligned}
\tag{13}
$$

and hence it accommodates only half of the westward interior transport (8).

For standard parameters, the frontal jet has an e-folding width of 40 km with a maximum speed of 3.6 m s$^{-1}$ and transport of 72 Sv, all of which are of the same order as the observed GSE [34]. As we shall see later, this transport would be boosted by the recirculation and the MOC to improve the observational comparison. It is important to note that since the frontal jet is fully specified by the prior constraints, its transport is independent of the basin width, in sharp contrast to the Sverdrup transport (Section 4).

### 3.3. EUC

Setting the stream function to zero at the outcrop, (7) and (13) imply that, in the interior,

$$
\psi = y - 1/2
\tag{14}
$$

The westward interior flow thus bifurcates at $y = 1/2$ (about 20° N) to form counter-rotating subtropical (anticyclonic) and tropical (cyclonic) gyres, the latter giving rise to the EUC. Physically, since the frontal jet along the outcrop is subjected to a local Coriolis parameter that is twice the interior average yet spans the same thermocline depth, it can only return half the interior transport, and the other half must return via the EUC. This bifurcation of the NEC is well observed in the western Pacific [27], with the northern and southern branches corresponding to the Kuroshio the Mindanao currents, respectively.

For the equatorial boundary layer where the EUC resides, the homogenized PV (2) and geostrophy (4) again combine to yield [35]

$$
u_{yy} - \varepsilon^{-2} y u = \varepsilon^{-1}
\tag{15}
$$

Defining a stretched coordinate $\varsigma \equiv \varepsilon^{-2/3} y$, then, to the accuracy of O $(\varepsilon^{1/3})$, (15) becomes

$$
u_{\varsigma\varsigma} - \varsigma u = 0
\tag{16}
$$

which has the solution

$$
u = C \cdot Ai \, (\varsigma)
\tag{17}
$$

$$
h = -\varepsilon^{1/3} C \cdot Ai'(\varsigma)
\tag{18}
$$

To determine the constant $C$, we note that the Bernoulli function

$$
B \equiv h + u^2/2
\tag{19}
$$

satisfies, subjected to (2), (4) and (6),

$$
B_\psi = 1
\tag{20}
$$

which yields

$$
B = 1/2 + \psi
\tag{21}
$$

given its value at the outcrop (from (11) and (12)). Since, at the equator, $\psi = 0$, hence $B = 1/2$, it yields $C \approx 2.41$. The solution for the equatorial boundary layer is shown in Figure 3, with its width magnified 5-fold.

Although the EUC was initially suggested to be driven by the easterly trade winds [36], its subtropical source is subsequently established from tracer observations [37,38] and numerical calculations [39]. Pedlosky [35] formulated an inertial model of the EUC, which, however, requires a prescription of the bifurcation latitude. In our formulation, the EUC

is an internal component of the gyre circulation hence fully specified by the prior constraints. Based on standard parameters, the model EUC would extend to about 2° N , as observed [40]. It has a speed of 3 m s$^{-1}$ and a hemispheric transport of 72 Sv (hence a total transport of 144 Sv), both are high compared with observations, which, however, would be greatly reduced by the presence of a tropical layer (not modelled here, see Section 1), as discussed next.

As seen in Figure 4, the tropical layer (polka-dotted) would depress the main thermocline downward as the warm layer thickness is constrained by the homogenized PV. Since the Bernoulli function at the equator, being set by its outcrop value, remains unchanged, one sees immediately that the EUC would be weakened. For a tropical layer thickness of 160 m (or non-dimensionally $h_s = .16$), the boundary condition to the EUC solution would be

$$h + u^2/2 = 1/2 - h_s = .34 \text{ at } \varsigma = 0 \tag{22}$$

which yields C = 1.76. The EUC thus is weakened from 3 to 2.3 m s$^{-1}$, a 23% reduction. As regards the EUC transport, a tropical layer extending to 10° N would reduce its subtropical supply by half from 72 to 36 Sv, and then the northern subsurface countercurrent (NSCC) would syphon another 10 Sv [40] to reduce this supply to 26 Sv, resulting in a total EUC transport of 52 Sv, which is now commensurate with its observed transport.

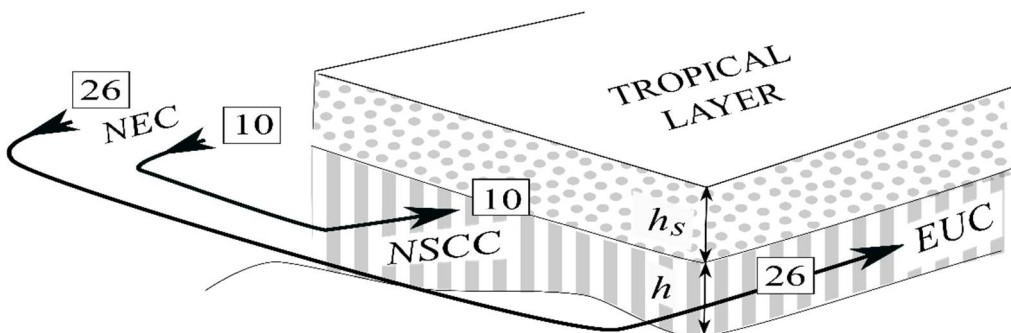

**Figure 4.** A schematic of the EUC when overlain by a tropical layer (polka-dotted). The tropical layer depresses the main thermocline to weaken the EUC via the Bernoulli law. In addition, the tropical layer encroaches on the NEC to reduce the subtropical supply of the EUC, which is also syphoned by the NSCC, resulting in a much-reduced EUC transport.

*3.4. MBC*

With the interior flow specified by the homogenized PV, it would drive meridional boundary currents by continuity. Along the western boundary, we define a stretch coordinate $\varsigma \equiv \varepsilon^{-1}x$, then the homogenized PV and geostrophy imply, respectively,

$$h = y + v_\varsigma \tag{23}$$

$$yv = h_\varsigma \tag{24}$$

It has the solution

$$v = A exp(-\sqrt{y}\varsigma) \tag{25}$$

$$h = y - \sqrt{y}A \, exp(-\sqrt{y}\varsigma) \tag{26}$$

where

$$A = \sqrt{y} - \sqrt{1-y} \tag{27}$$

by applying (21) and $\psi = 0$ along x = 0.

The solution is shown in Figure 3 in dashed lines, noting that the meridional flow is of the opposite sign on two sides of the bifurcation, and they are reversed along the eastern boundary. Unlike the interior thermocline that deepens monotonically with the latitude until it surfaces abruptly across a narrow frontal jet, the thermocline along the western boundary shoals away from its maximum mid-latitude depth. Since the boundary layer is

scaled by the deformation radius, it narrows with the latitude, and the southern branch has a deeper thermocline at the coast than the interior; both these features are consistent with observations [27] (their Figure 10).

In contrast to the laminar dynamics, the eastern boundary current (EBC) is not prohibited, which is more compliant with its observed prominence [41]. In fact, based on the laminar dynamics, any outcrop intersecting the eastern boundary would remain at the surface throughout the subtropics, a singularity unobserved in the ocean. In an eddying ocean, the PV homogenization has voided the laminar constraint, so the thermocline depth simply counter-varies with the velocity, and both are allowed.

### 3.5. Recirculation

The basic physics of the recirculation can be gleaned from Figure 1. Because of the homogenized hence finite PV, the western boundary current may depart the boundary only in a tangent, which then curls increasingly negatively due to the beta effect until it turns south, following which the trend reverses, resulting in a meandering path. Because of the invariable dissipation and blurring of the thermal front, the initial arc is the most sharply defined. Within this arc, the homogenized PV would continue to deepen the thermocline accompanied by westward geostrophic flow, just like the subtropical interior—except the westward flow is now blocked by the arc to recirculate through the frontal jet, thus enhancing its maximum transport at the apex.

Since the recirculation is couched within the arc, we need to determine its trajectory. In [42], we solved a similar problem of the retroflected Agulhas Current; the only difference in the present formulation is that the far-field thermocline depth of the frontal jet is not constant but increases with latitude. As their approximations of "narrow jet" and "small meander amplitude" remain valid, their solution of the trajectory is unchanged hence need not be repeated here. A key result is the meander amplitude $l_a$ given by [42] (their Eq. 5.5),

$$l_a = \left( \frac{2M}{\beta Q} \right)^{1/2} \tag{28}$$

where $Q$ and $M$ are the mass and momentum fluxes, respectively, of the frontal jet at the separation point. Applying the solution (11) and (12), we derive

$$l_a \approx 0.82 \, (r_c l)^{1/2} \tag{29}$$

or the meander amplitude is roughly the geometric mean of the deformation radius and the warm-layer extent. Applying the standard values, the meander amplitude is 400 km, not unlike that observed. As the recirculation takes on the PV of the warm layer and is framed within the arc, itself a function only of this PV, the recirculation is an integral part of the subtropical gyre, and hence is fully specified by the prior constraints. This is to be contrasted with previous models of the recirculation, which need to prescribe its PV and the spatial confinement [43].

The framing of our recirculation within the warm arc of the GSE is consistent with observations [44]. The deduced transport increase is due solely to the deepening thermocline while the jet speed remains unchanged on account of (21), as indeed the observed case [34]. For an arc extending 330 km beyond the separation latitude (29), the jet transport would peak at 84 Sv, 30% of which resides in the recirculation, supporting its well-subscribed importance [45].

### 3.6. MOC

The MOC transport has been previously derived in our climate theory [7], which constitutes an integral part of the GOC. The MOC outflow consists of both the mean flow and the shedding of warm eddies [13], which returns to the warm layer via the deep western boundary current and upwelling, and as cold eddies absorbed across the subtropical front. Since the upwelling occurs mainly in the western boundary layer [2], it

contributes to the northward increase in the GS transport. Together with the geographical constraints that cold eddies are absorbed near the separation point and warm eddies are not shed until after the initial arc [46], the MOC would add to the maximum transport of the GSE, accentuating the western intensification. Quantitatively, a MOC of 20 Sv [47] would boost the GSE transport from 84 to 104 Sv, further improving its observational comparison.

## 4. Sverdrup Perspective

The Sverdrup balance is based on laminar dynamics as it predates the satellite observation of teeming eddies and their well-demonstrated importance in mixing the PV. Since the Sverdrup velocity is too small (several mm/s) to be detected amid the eddy motion, the observational support of the Sverdrup balance is derived mainly from its implied transport, which typically involves adjusting the level-of-no-motion or integration depth to produce a match [48,49]. However, even if the Sverdrup transport may account for the extratropical GS [50], the agreement is necessarily fortuitous since the Sverdrup transport begins to decrease beyond about 25° N while the GS transport increases unabated to its northernmost reach. Equally as seriously, the Sverdrup transport, being proportional to the basin width, would double in the North Pacific compared with the North Atlantic [51], yet the observed Kuroshio and GS have similar transports [52].

In contrast to the Sverdrup balance, the homogenized PV has tangible observational supports, as detailed in Section 2, and it readily resolves the above transport shortcomings: the northward deepening of the thermocline would continually augment the GS transport; and since the GSE transport is specified by meridional thermal properties, it is independent of the basin width.

While coarse-grained numerical calculations have validated the Sverdrup dynamics, they provide poor simulations of the observed GOC [53]. The simulations are much improved when the eddies are resolved [54], which however are at the expense of a shrinking Sverdrup domain [5,55]; and then the laminar dynamics has difficulty in accommodating a subtropical front to produce a closed circulation [56]. These deficiencies suggest that although the Sverdrup balance may hold locally, it falls short as an organization principle of the GOC.

Yet, the Sverdrup dynamics remains widely ascribed in wind-driven theories, which can be attributed to its simplicity in explaining the western intensification and a dearth of alternative theories that have produced closed circulation when the eddy mixing of PV is of the first-order importance. The present theory can fill this void: by considering the asymptotic limit of homogenized PV, it allows an analytical solution of closed circulation, which moreover is inherently western intensified—the Sverdrup dynamics thus need not be the sole progenitor of the western intensification, as commonly perceived.

Since the upper-bound GOC is an asymptotic state, one expects the observed GOC to retain remnants of the Sverdrup dynamics, but with the PV homogenization representing a higher order balance, the Sverdrup remnant constitutes only a regular perturbation, which would not singularly perturb the upper-bound GOC. To produce a more refined GOC, one may simply take the Sverdrup solution (for example, [33]) adjusted for the potency factor (Section 2) and apply it to the upper-bound GOC. For a potency factor of 0.2, the Sverdrup remnant would augment the extratropical GS transport by about 10 Sv but has little effect on the GSE. More noticeably, it would tilt the interior thermocline downward to the west (by several tens of meters), an observational feature that is absent from the upper-bound GOC. Clearly, the Sverdrup remnant is needed for a more detailed explanation of the GOC, but we have nonetheless demonstrated that the upper-bound GOC, which contains no wind curl or vestige of the Sverdrup balance, can explain the broad pattern of the observed GOC. The GOC simulated by primitive-equation models naturally has retained the Sverdrup remnant, but a recent eddy-resolving experiment of zero wind [3] suggests that Sverdrup dynamics is not needed to produce a subtropical gyre, a tantalizing claim that is in fact aligned with our view, but with an important caveat. That is, our upper-bound GOC varies linearly with the mean thermocline depth, which can be regulated by the wind

via the mechanical energy balance (Section 2), so although PV mixing short-circuits the wind curl, one still needs strong enough wind to drive the observed GOC. In the absence of such wind, [3] was compelled to artificially raise the vertical diffusivity to deepen the thermocline and strengthen the gyre.

## 5. Conclusions

This paper completes our five-part climate theory aiming to derive the generic climate state forced only by the solar insolation [6–9]. Subjected to the bulk thermal properties previously determined, we derive the upper-bound GOC above the main thermocline when PV is homogenized, the latter being supported on theoretical, computational, and observational grounds. While the upper-bound GOC represents an asymptotic limit, it has nonetheless replicated broad features of the observed GOC, including a western-intensified subtropical gyre and a cyclonic tropical gyre feeding the EUC.

Since PV homogenization has short-circuited the wind curl to fully remove the vestige of the Sverdrup balance, the latter need not be the sole progenitor of the western intensification, as commonly perceived. In addition, the Sverdrup dynamics has encountered singularity at the subtropical front in producing a closed circulation, a singularity that is readily removed by the upper-bound GOC. Since PV homogenization retains higher order PV balance than the Sverdrup dynamics, the remnant of the latter constitutes merely a regular perturbation, which nonetheless is needed to explain the finer structure of the observed GOC.

In a companion paper on the general atmosphere circulation [9], we have shown that its upper bound may replicate the observed prevailing wind as well. Together with the present paper, we posit that PV homogenization provides a unifying dynamical principle for the general planetary circulations, which may be interpreted as the maximum macroscopic motion extractable by microscopic stirring—within the confines of thermal differentiation.

**Funding:** This research received no external funding.

**Institutional Review Board Statement:** Not applicable.

**Informed Consent Statement:** Not applicable.

**Acknowledgments:** I have benefitted from discussions with Dake Chen, Ryan Abernathey and Tongya Liu.

**Conflicts of Interest:** The author declares no conflict of interest.

## Appendix A

| | |
|---|---|
| EBC | Eastern boundary current |
| EUC | Equatorial undercurrent |
| GAC | General atmosphere circulation |
| GOC | General ocean circulation |
| GS | Gulf Stream |
| GSE | Gulf Stream extension |
| MBC | Meridional boundary current |
| MEP | Maximum entropy production |
| MOC | Meridional overturning circulation |
| NEC | North equatorial current |
| NSCC | Northern subsurface countercurrent |
| NT | Nonequilibrium thermodynamics |
| PV | Potential vorticity |
| $B$ | Bernoulli function |
| $g'$ | Reduced gravity $(= 1.3 \times 10^{-2} \text{ m s}^{-2})$ |

| | |
|---|---|
| $h$ | Thermocline depth |
| $\bar{h}$ | Mean thermocline depth (=.5 km) |
| $[h]$ | Thermocline depth scale (=$2\bar{h}$ = 1 km) |
| $l$ | Warm-layer extent (=4000 km) |
| $M$ | GS momentum flux at separation |
| $P$ | Columnar PV |
| $[P]$ | PV scale (= $\beta l / [h]$) |
| $Q$ | GS mass transport at separation |
| $r_C$ | Deformation radius (= $(g'[h])^{1/2}(\beta l)^{-1}$ = 40 km) |
| $u$ | Zonal current |
| $[u]$ | Velocity scale (=$(g'[h])^{1/2}$ = 3.6 m s$^{-1}$) |
| $v$ | Meridional current |
| $y$ | Latitudinal distance |
| $[y]$ | Distance scale (= $l$ = 4000 km) |
| $\beta$ | Gradient of Coriolis parameter (= $2 \times 10^{-11}$ m$^{-1}$ s$^{-1}$) |
| $\varepsilon$ | $\equiv r_C / l = 0.01$ |
| $[\psi]$ | Transport scale (= $[h][u]r_c$ = 144 Sv) |

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
