# Peer review of "Upper-Bound General Circulation of the Ocean: A Theoretical Exposition"

_jmse, doi:10.3390/jmse9101090_

Round 1

Reviewer 1 Report

This paper is within the scope of the JMSE. The title of the manuscript is interesting. Generally speaking, the paper is well written and the language is very concise. I  suggest a minor revision for this manuscript before publication:

1. The authors are required to emphasize their unique findings derived from the present study.

2. Is the potential vorticity homogenized in the upper-bound general circulations? More details should be provided in the manuscript.

3. I suggest adding some documents from the past three years in the introduction section. 

Author Response

I want to thank the reviewer for the highly constructive comments.  My responses are given below in Italics.

This paper is within the scope of the JMSE. The title of the manuscript is interesting. Generally speaking, the paper is well written and the language is very concise. I  suggest a minor revision for this manuscript before publication:

  1. The authors are required to emphasize their unique findings derived from the present study.

I have modified the abstract and expanded the conclusions (Section 5) to highlight the unique findings of the present study.   

  1. Is the potential vorticity homogenized in the upper-bound general circulations? More details should be provided in the manuscript.

I have discussed in Section 2 theoretical, computational, and observational grounds of homogenized PV as relevant to the observed ocean.  I have also slightly modified the discussion in Section 3.1 as to why the homogenized PV, by maximizing the velocity scale, renders the upper-bound GOC.   

  1. I suggest adding some documents from the past three years in the introduction section.

I have found only two relevant papers from the last three years [3] and [5], which are cited.

Reviewer 2 Report

This paper discusses the upper layer circulation on the basis of the potential vorticity conservation law.
However, it is difficult to understand the assumption of the formula.
For example, does Equation (1) assume the equatorial beta plane?
In that case, there is a problem with the validity of the argument in this paper.
Or is Equation (1) the potential vorticity in the beta plane approximation,
and subtracting the constant?
It is not valid unless h is a constant.

The basic equations, such as potential vorticity conservation, should be written according to standard textbooks.
Then you need to describe what approximations or normalizations you are using.

Author Response

This paper discusses the upper layer circulation on the basis of the potential vorticity conservation law.
However, it is difficult to understand the assumption of the formula.
For example, does Equation (1) assume the equatorial beta plane?
In that case, there is a problem with the validity of the argument in this paper.
Or is Equation (1) the potential vorticity in the beta plane approximation,
and subtracting the constant?
It is not valid unless h is a constant.

The basic equations, such as potential vorticity conservation, should be written according to standard textbooks.
Then you need to describe what approximations or normalizations you are using.

  • I have rewritten the beginning of Section 3 where the PV is written out in its traditional dimensional form (from textbooks) and justified the equatorial beta-plane approximation (noting that the thermocline depth is not limited to a small perturbation about the mean, as in quasi-geostrophic models). I then define the scales for the model variables, which appear only in dimensionless form thereafter. 

Round 2

Reviewer 2 Report

The responses are adequate and it is possible to publish.

Fig.1, MBC is not used in the text.

It seems to be meridional  boundary currents (Line 203)

Author Response

I have corrected the errors of Fig. 1 and MBC, which have consistent usages now.

thank you for your attention to details.